# Can 12-Week Resistance Training Improve Muscle Strength, Dynamic Balance and the Metabolic Profile in Older Adults with Type 2 Diabetes Mellitus?

**DOI:** 10.3390/ijerph22020184

**Published:** 2025-01-28

**Authors:** André Luiz de Seixas Soares, Guilherme Carlos Brech, Adriana Machado-Lima, Joselma Rodrigues dos Santos, Júlia Maria D’ Andréa Greve, Marcus Vinicius Grecco, Mara Afonso, Juliana Cristina Sousa, Ariana Tito Rodrigues, Matheus Henrique dos Santos Lino, Vanderlei Carneiro da Silva, Patricia Nemara Freitas de Souza Carneiro, Alexandre Lopes Evangelista, Catherine L. Davis, Angelica Castilho Alonso

**Affiliations:** 1Graduate Program in Aging Sciences, University Sao Judas Tadeu (USJT), São Paulo CEP 03166-000, SP, Brazil; andre.suporte@hotmail.com (A.L.d.S.S.); nutriarianatito@gmail.com (A.T.R.); matheushsl2001@gmail.com (M.H.d.S.L.); vnd.cs@hotmail.com (V.C.d.S.); patricianemara@hotmail.com (P.N.F.d.S.C.); angelicacastilho@msn.com (A.C.A.); 2Georgia Prevention Institute, Medical College of Georgia, Augusta University, Augusta, GA 30912, USA; katie.davis@augusta.edu; 3Laboratory Study of Movement, Instituto de Ortopedia e Traumatologia do Hospital das Clínicas (IOT-HC), Faculdade de Medicina FMUSP, Universidade de São Paulo, São Paulo CEP 05402-000, SP, Brazil; joselma.rodrigues@hotmail.com (J.R.d.S.); jgreve@usp.br (J.M.D.A.G.); mvgrecco@ig.com.br (M.V.G.); mara.afonso@hc.fm.usp.br (M.A.); juliana.csousa@hc.fm.usp.br (J.C.S.); contato@alexandrelevangelista.com.br (A.L.E.)

**Keywords:** diabetes mellitus, glycemia, cardiovascular risk, older adults, resistance exercise

## Abstract

The present study aimed to evaluate the effects of 12-week resistance training (RT) on muscle strength, dynamic balance, glycemic control and the lipid profile. Methods: The Laboratory of Movement Studies in the University of São Paulo, Brazil, developed this longitudinal study between 2021 and 2023. It assessed 62 males with type 2 diabetes mellitus pre and post an RT protocol. The participants, who were 69.8 (±3.9) years old, took part in a 12-week twice-weekly RT program. Three sets of eight to twelve repetitions each were executed for eight exercises targeting the large muscle groups. The intensity was set between 7 and 8 out of 10 for perceived effort, according to the Omni Resistance Exercise Scale. All participants were evaluated pre and post in knee extensor and flexor strength by isokinetic dynamometry, handgrip strength by manual dynamometry and dynamic postural balance by a force platform, as well as blood tests to determine the lipid and glycemic profiles. For comparison, paired t or Wilcoxon tests were used at an alpha of 5%. Results: There was an improvement in muscular strength by handgrip restricted to the non-dominant side (*p* = 0.033) and for the bilateral knee flexors (*p* < 0.001) and extensors (*p* < 0.001), as determined by isokinetic dynamometry. There was no improvement in dynamic postural balance, glycemic control or lipid control. Conclusions: The 12-week RT promoted improved muscle strength in knee extension and flexion and non-dominant grip pressure but did not affect dynamic balance, glycemic control or the lipid profile.

## 1. Introduction

The prevalence of type 2 diabetes mellitus (T2DM) is expected to continue to rise globally, with an estimated 783 million people (12.2% of the world’s population) projected to live with T2DM by 2045. The prevalence in older adults is also expected to increase significantly, with an estimated 276.2 million people (35.3% of the total number of cases) being in this age group by 2045 [1,2,3].

This increasing prevalence of T2DM is a major public health concern and calls for effective prevention and management strategies. It is essential to raise awareness about the risk factors associated with T2DM and promote lifestyle changes, such as regular physical activity, healthy eating habits and weight management, to prevent or delay the onset of T2DM. Effective management of T2DM also requires early diagnosis, regular monitoring and appropriate medical care to forestall its complications [1,4].

In addition to the natural declines that occur with aging, hyperglycemia, insulin resistance and dyslipidemia are distinct yet interconnected factors that aggravate neural and muscular impairments characteristic of the pathophysiology of T2DM [5]. Changes in neural and muscular function may lead to ineffective motor efferent function in peripheral neural tissues that are responsible for intra- and intermuscular coordination, leading to impaired strength and motor skills [5,6]. Similarly, defective sensory afferent pathways fail to provide crucial inputs to the central nervous system for balance processing and proper activation of these already impaired efferent pathways [6]. Beyond that, older adults with T2DM have deficits in postural balance, especially in situations that prioritize the motor and visual systems, also commonly impaired in this population [7]. Hence, we can affirm that the afferent neurons, central processing and efferent neurons are impaired by direct mechanisms caused by high levels of glycemia, insulinemia and dyslipidemia.

The supply of oxygen and nutrients is potentially harmed by micro- and macrovascular disease, both caused by persistent hyperglycemia [8]. This condition combines with the mentioned direct peripheral nerve damage, and it can worsen peripheral and central nervous systems in their morphology and functionality too [9].

Impairments in muscular function are not restricted to their interactions with neural signaling or lack of nutrients and oxygen. Dysglycemia causes losses in the protein anabolic pathways simultaneously with the exacerbation of its catabolic cascades. Similarly, elevated contents of glucose and lipids also lead to mitochondrial dysfunction, inflammation, oxidative stress and cell death [10,11].

Physiological changes associated with or preceding the development of T2DM increase susceptibility to neuromuscular dysfunction, often linked to retinopathy and vestibulopathy. Additionally, peripheral neuropathy, especially in the feet and toes, can impair balance and significantly elevate the risk of falls due to reduced sensory perception [7,12]. This combination substantially diminishes physical function, thereby increasing the likelihood of falls [12,13]. Therefore, personalized exercise programs are essential to enhance physical abilities and prevent falls in this population.

Resistance training (RT) is an exercise category recognized as a therapeutic intervention in glycemic control, the metabolic profile, strength, muscle mass gain and, consequently, physical functionality in older adults with T2DM [14]. The training strategy has been proven to be a key component of health management in this population [15]. It can also increase the quality of life, in which independence and physical autonomy are crucial factors [16,17,18].

Falls represent a significant health risk for older adults, especially those with T2DM [2,7]. Changes in lower-limb muscle strength and postural balance are among the main intrinsic risk factors associated with falls [7,12]. While RT is effective in strengthening antigravity muscles, postural balance requires the integration of multiple systems, such as the visual, proprioceptive and vestibular systems [7]. These systems, due to persistent hyperglycemia associated with T2DM and the aging process, experience more accelerated functional decline, impairing the central nervous system’s ability to integrate sensory information and execute proper musculoskeletal responses, thereby increasing vulnerability to falls [2,7,12,13]. Better understanding the relationship between muscle strength and postural balance is essential for developing more effective interventions aimed at reducing the risk of falls and improving the quality of life for this population [7].

Evangelista et al. [19] affirm that although muscle strength and postural balance are physical skills that rely on independent neuromuscular components, they complement each other in enabling the performance of daily and sporting activities. However, the interrelationship between these skills is still not fully understood, suggesting that further investigations are needed to clarify the discrepancies and optimize intervention strategies [19].

The main focus of studies on T2DM and RT has been glycemic control, a key aspect in managing the disease [15,20]. While many studies have explored the effects of RT on muscle strength [10,14], there is a notable lack of research simultaneously assessing its impacts on dynamic balance using standardized and high-precision tools [7]. Our study aimed to address this gap by providing an integrated analysis of how RT can influence both muscular and postural parameters in this specific population. Additionally, we investigated other relevant health aspects, such as lipid and glycemic profiles, broadening the understanding of RT’s benefits for individuals with T2DM.

The present study evaluated how a 12-week RT program affects lipid profiles, glycemic control, dynamic balance and musculature in older men with T2DM.

Our hypothesis was that a 12-week RT program would significantly improve muscle strength and glycemic and lipid profiles, but would not improve postural balance.

## 2. Materials and Methods

### 2.1. Experimental Design, Location and Ethics

This study was performed at the Motion Study Laboratory of the Institute of Orthopedics and Traumatology at the University Hospital of São Paulo School of Medicine (HC-FMUSP) in partnership with the São Judas Tadeu University (USJT), with approval granted by the Ethics Committee of the University of São Paulo (CAAE: 39202214.8.0000.0065).

### 2.2. Subjects

In this experimental study (pre and post), 62 men with T2DM between the ages of 65 and 79 with BMIs between 22 and 32 kg/m^2^ and glycated hemoglobin (HbA1c) ranging between 6 and 9% were included. The patients had taken a stable dose of medicine for three or more months (insulin, oral anti-diabetic or a combination of both). Glomerular filtration rate (GFR) estimation, using the Modification of Diet in Renal Disease (MDRD) method, was used to assess renal function, which had to be greater than 60 mL/h. Serum values of the liver enzymes aspartate aminotransferase (AST) and alanine aminotransferase (ALT) could not be more than 2.5 times the upper limit. Additionally, they had to be free of any musculoskeletal pain, incapacitating motor disorders or cardiovascular or musculoskeletal conditions that could have prevented them from performing RT. Furthermore, they could not have participated in any RT practice in the three months before the beginning of the pre-intervention tests. Participants with more than three absences during the training protocol or any adverse event were excluded during the time of the intervention. Eight participants were excluded, two for traveling during the RT, four for more than three absences and two for health issues. The sample size was determined using data from a prior study that our research team carried out. The outcome measure was HbA1c, and the test power was 90% with an alpha value (probability of type I error) set at 5% and a beta value (probability of type II error) at 90%. Therefore, we required 62 participants in the study group.

### 2.3. Procedures

The subjects were recruited by advertising at local health centers and on the hospital’s Facebook. Each potential participant who accepted the invitation and agreed to visit the facility was then given a thorough explanation of the research. The informed consent form was signed by those who decided to participate. After that, appointments were made with the clinician, who assessed the medical conditions and gave the clearance for the completion of all pre-study evaluations, laboratory testing and blood exams. The data collection took place between August 2021 and December 2023. The tests were performed by physiotherapists trained and experienced in applying them.

### 2.4. Instruments

#### Muscle Strength Assessment

The handgrip strength (HGS) test was performed using a Jamar hand dynamometer, measured in kilograms/force (kg/f). Participants were instructed to hold the dynamometer while sitting in a chair without arms, with their elbows at a 90-degree angle and forearms parallel to the floor. After that, they were given standardized instructions and told to squeeze as hard as they could. To prevent muscle fatigue, tests were alternately performed on the right and left hand, followed by a one-minute break. For the analysis, the average of the three measurements per side was utilized [21].

The maximum strength of each participant was assessed using isokinetic dynamometry with the Biodex^®^ Multi-joint System 3 (Biodex Medical™, Shirley, NY, USA). The procedure began with a standardized warm-up. Participants were then positioned in the dynamometer with their hips flexed at 90 degrees and securely strapped to the chair to maintain proper alignment. The evaluation focused on concentric movements for knee joint extension and flexion, starting with the dominant limb at a velocity of 60°/s. To ensure participants were familiar with the equipment, each individual performed four submaximal repetitions as a familiarization exercise. After a 60 s rest, participants executed two sets of five repetitions of knee extension and flexion, applying the maximum effort during each repetition. For data analysis on the impact of motor learning on clinical isokinetic performance, results from the second set were used. Throughout the test, we provided consistent, standardized verbal encouragement to ensure that the participants exerted their highest effort during contractions. This is crucial in isokinetic tests because the equipment is programmed to move at a specific angular speed. The machine automatically adjusts resistance to maintain the programmed speed of 60° per second, regardless of the force applied by the participant. The equipment processes data obtained from measuring the strength of each part of the movement arc to produce results [22,23].

### 2.5. Dynamic Balance Assessment

The NeuroCom Balance Master^®^ force platform system (NeuroCom International, Inc., Clackamas, OR, USA), which consists of a computer with a force platform that collects data using piezoelectric crystal transducers, was used to evaluate the participants before and after the intervention. At a sampling rate of 100 Hz, the force-platform data included the X (0.08 cm) and Y (0.25 cm) positions of the vertical force center and the total vertical force (0.1 N). With this technology, the transducers send pressure inputs to the computer every 10 ms. After that, the computer determines the person’s dynamic center of gravity and measures their sway velocity (degrees/second) over a predetermined period of time to assess postural stability. The same evaluator conducted each test, and the places and sequence of the testing positions were standardized. This study made use of the Step/Quick Turn, Sit-to-Stand Transfer Test and Step Up and Over Test Protocols, which are well correlated with the risk of falls and have good similarities with daily mobility demands [7,23].

Sit-to-Stand Test: The participant was instructed to sit on a bench without a backrest, with their feet apart and knees bent at a 90-degree angle. They were then asked to stand up quickly and safely, remaining standing for a few seconds. This test was repeated three times with an interval of 30 s between each attempt. The measured parameters included weight transfer (as a percentage of body mass) and center of gravity sway (°/s) (during the standing motion) [7,23].

Step/Quick Turn: Each participant was instructed beforehand to walk onto the platform, turn 180 degrees and return to the starting point. The tests commenced first on the left side and then on the right side, with each side being repeated three times, allowing for a 30 s interval between each repetition. The parameters analyzed were the turn time (time to complete the task in seconds) and turn sway (balance while performing the turn, in °/s) [7,23].

Step Up/Over: To conduct the test, a 20 cm step was positioned in front of each participant. Participants were instructed by the evaluator to step onto the platform using their left leg while maintaining an upright posture. They then stepped over the platform with their right leg and concluded the movement by stepping down with their left leg onto the floor. The analyzed parameters were the lift up (% weight), movement time (time required to perform the task) and impact (% weight) [7,23].

### 2.6. Biochemical Profile

Participants who were clinically approved underwent blood sample collection for laboratory analyses. Peripheral blood samples (20 mL) were collected in tubes containing EDTA anticoagulant after a 12 h fasting period, at baseline and at the end of the 12-week exercise protocol. Whole blood with EDTA was centrifuged to obtain plasma. The circulating concentration of glucose, insulin, fructosamine, hepatic enzymes and glycated hemoglobin (HbA1c) were assayed, and insulin resistance was determined by the Homeostatic Model Assessment for Insulin Resistance (HOMA-IR) formula. Score = (Fasting insulin, uIU/mL) × (Fasting glucose, mg/dL)/405 [24]. The lipid profile including triglycerides, very low-density lipoprotein (VLDL), high-density lipoprotein (HDL), low-density lipoprotein (LDL) and total cholesterol was measured (Labtest, Minas Gerais, Brazil).

### 2.7. Exercise Protocol

For 12 weeks, the exercise routine was conducted twice a week on Tuesdays and Thursdays in the mornings. The sessions were approximately 40 min in the Movement Study Laboratory. All exercises were supervised by trained and qualified physical education professionals. The exercises chosen were for the large muscle groups. The leg press, leg extension, seated row, leg curl, arm curl, calf raise and shoulder press were performed [23]. Before the main part of the session began, we conducted a warm-up period lasting 5 to 10 min. During this time, we performed exercises using the same equipment that would be used in the training, at 50% of the intensity and load that participants would use during the actual training. After that, three sets of 8 to 12 repetitions were carried out for each exercise. According to the OMNI-RE scale, which ranges from 0 to 10—where 0 represents “no effort” and 10 represents “maximum effort”—the intensity of the activity was rated between 7 and 8. This self-assessment tool allows individuals to evaluate their perceived effort during physical activity, making it especially useful for monitoring and adjusting exercise intensity. The OMNI scale is regarded as a reliable tool for intensity management, even among older adults. Additionally, there were one to two minutes of breaks between workout sets [24,25].

### 2.8. Statistical Analysis

Data were stored in the IBM SPSS Statistics 24 software and presented using means, standard deviations and medians. The Shapiro–Wilk test was performed to verify whether the variables were normally distributed. The data were analyzed by comparing pre- and post-intervention results. We adopted the paired *t*-test or, when the variable did not meet normality, the Wilcoxon signed-rank test. A significance level of 5% was used for the entire analysis.

## 3. Results

The characterization of the participants is described in Table 1.

There was a significant gain between pre- and post-exercise in the non-dominant HGS and in quadriceps and hamstring muscle strength evaluated by isokinetic dynamometry (Table 2).

There were no significant differences in dynamic balance variables evaluated by the force platform (Table 3).

There was no difference in the pre- and post-glycemic control: HbA1c, glucose, fructosamine and homeostatic model assessment for insulin resistance (HOMA IR) (Figure 1).

There was no difference between pre and post lipid profiles (Figure 2).

## 4. Discussion

This study’s main findings indicated that 12 weeks of progressive RT significantly improved muscle strength in older male adults with T2DM. This training method systematically increased the loads over time, aiming to maintain an exercise intensity between 7 and 8 on the OMNI scale as participants gained strength. The training effectively stimulated muscle fibers, enhancing strength and improving metabolic health. However, no clinically significant differences were observed in dynamic balance, glycemic control or lipid profiles. It is well-known that T2DM significantly increases the risk of developing sarcopenia, physical disability and falls [26,27].

Muscle mass declines at a rate of 3% per year in healthy individuals over the age of 60 [19]. However, in individuals with T2DM, the loss of skeletal muscle mass occurs 10% to 20% faster in men and can be up to 100% faster in women compared to healthy older adults [7,23]. The current study demonstrated that the exercise protocol not only halted the loss of strength and muscle mass but also led to an increase in muscle strength. This was evidenced by improvements in knee extensor strength, as measured by isokinetic dynamometry, and HGS.

The notable improvement in HGS on the non-dominant side compared to the dominant side, as assessed by the HGS test, may be attributed to the neural aspects of muscle function. It is common to observe differences in muscle strength, with the dominant side often being up to 10% stronger. This disparity can be explained by the lower demand for strength and motor coordination activities on the non-dominant side [28,29]. Similar differences were also observed in the population studied during the baseline assessment. The intervention decreased the differences without implementing a specific compensation strategy. This suggests that the stimuli may not have been strong enough to produce further adaptations in the better-conditioned side, but they were sufficient to impact the less-conditioned neuromuscular mechanisms on the opposite side.

Considering that the magnitude of improvements in functional responses is inversely proportional to its level before the intervention [30], we can assume that, if our population had lower levels of muscular strength or physical function, we might have seen more significant improvements on both sides, following the same protocol. Conversely, if the protocol were longer, more intense or even had a higher volume of exercises, we possibly could have obtained more significant increases on both sides.

The increase in muscle strength did not lead to a significant improvement in dynamic balance from a clinical standpoint. It was anticipated that older adults, including those with T2DM, would show improved scores and significant changes in functional tasks after participating in an RT program and gaining strength. This expectation aligns with several related studies that have linked muscle strength with improvements in functional tasks [23,31,32]. We also observed that they did not have any regression in balance or physical functionality tests between the baseline and post-intervention period. These findings are encouraging, especially considering that declines in these abilities are commonly noted in studies involving older adults, even those without T2DM [7,33,34].

Strength training interventions are not directly associated with better performance in static or reactive postural control [35]. Also, vestibular and proprioceptive information are key components in amplifying the translation of strength to dynamic balance performance. As aforementioned, common characteristics of people with T2DM are central and peripheral nerve damage, as well as vestibulopathy and retinopathy, which can impair physical functionality and balance. We can consider these factors potential obstacles to physical functional gains in RT alone. According to several studies [31,33,34,36,37], a multimodal program could be more effective in improving each of the postural control components and their interactions, which are essential for better performance in functional tasks.

Our data did not show significant reductions in fasting serum glucose levels, fructosamine (which reflects the average blood glucose over the last 2 to 3 weeks) or HbA1c, a marker considered the gold standard in glycemic control and for measuring blood glucose levels over a 12-week period [38]. These results contrast with evidence from the literature, such as a meta-analysis of 24 clinical trials, which indicates that RT reduces HbA1c values and contributes to the reduction in serum glucose concentrations [39]. Even in the presence of insulin resistance, muscle contraction induced by exercise enhances glucose utilization through increased blood flow, and in the long term, improves mitochondrial function [18]. These processes promote greater glucose uptake from the bloodstream, thus reducing hyperinsulinemia [39]. However, it is important to note that our participants started this study with HbA1c within the range considered controlled by the SBD [40], and the RT helped maintain this good control. Therefore, we can infer that increasing the duration and/or volume of training sessions could provide additional benefits.

In our study, we did not observe any significant changes in the lipid profile. Nonetheless, the serum concentrations of the evaluated markers were already within acceptable values, which can potentially inhibit additional changes, as seen in different non-pharmacological approaches with this purpose [41]. Additionally, the reduction in HbA1C can have long-term benefits for managing lipid levels by improving glycemic control. As insulin resistance decreases, so does the production of VLDL in the liver, resulting in lower levels of LDL and triglycerides in the bloodstream [42]. Moreover, engaging in long-term physical exercise enhances mitochondrial oxidative capacity, which results in increased uptake of free fatty acids in muscle tissue. This in turn improves lipidemia and systemic inflammation, both of which are common in older adults with T2DM [43].

A systematic review with network meta-analysis carried out by Pan et al. [44] aimed to evaluate the impact of different exercises on glycemic control, cardiovascular risk and decrease in body mass in patients with T2DM, totaling 37 studies with 2208 patients. Of these, 26 studies with 1729 participants found that aerobic, resistance and combined supervised exercises induced a decrease in glycated hemoglobin (HbA1c) and fasting glucose compared to control groups with unsupervised exercises or without exercises. Regarding cardiac risk, 22 studies were included with 1323 participants, finding that aerobic, resistance and combined supervised exercises improved the lipid profile, and that specifically RT improved the values of HDL, LDL and total cholesterol. Thus, we can see the propensity to improve the lipid profile as a desired outcome, though this requires a long exercise training period in order to produce substantial changes in its values, as corroborating Pan et al. [44].

This study had limitations due to the absence of a control group. The diversity of subclinical conditions in T2DM populations makes sample stratification complex. Nonetheless, it is still crucial to clarify the optimal relationships between the volume, intensity and specificity of physical training. These factors are capable of promoting morphological changes and metabolic and functional adaptations, effectively enhancing independence and autonomy and reducing the risk of falls in older adults with T2DM.

The clinical implications of the present study lie in improving muscle strength and glycemic control. Furthermore, this study indicates that the effectiveness of balance improvement interventions depends on using tailored strategies based on individuals’ starting points. Therefore, longer and more comprehensive interventions may be necessary to achieve significant improvements in the metabolic profile.

## 5. Conclusions

The present study demonstrates that a 12-week RT program improves muscle strength in knee extensors and flexors and non-dominant handgrip strength in older adults with T2DM. However, no significant effects were observed on dynamic balance, glycemic control or lipid profiles. These findings suggest that while RT is effective for enhancing muscle strength, additional interventions may be required to address broader health outcomes. Future research should investigate combined training modalities and optimized protocols to maximize functional and metabolic benefits in this population.

## Figures and Tables

**Figure 1 ijerph-22-00184-f001:**
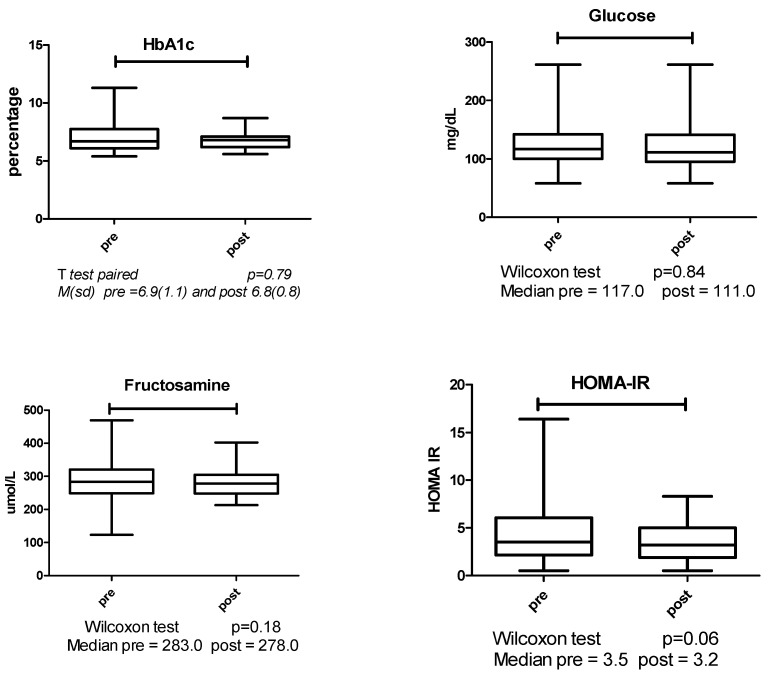
Comparison of pre- and post-exercise glycemic control.

**Figure 2 ijerph-22-00184-f002:**
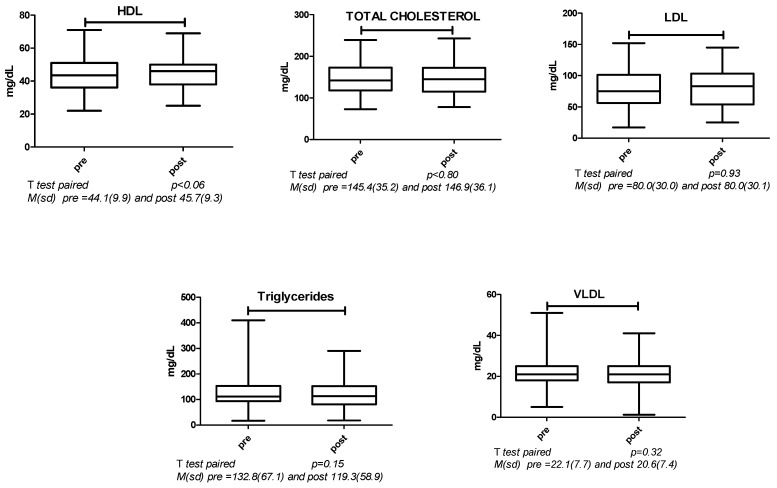
Comparison of pre- and post-exercise lipid profiles.

**Table 1 ijerph-22-00184-t001:** Characterization of the participants with T2DM, São Paulo, Brazil, 2021–2023, N = 62.

		N (%)	
Falls in the Last Year			
No		48 (77)	
Yes		14 (23)	
Smoking			
Ex-smokers		25 (40)	
No		33 (53)	
Yes		4 (6.5)	
Alcohol Consumption			
Social Drinking		31(50)	
No		26 (42)	
Daily		5 (8)	
Ethnicity			
Caucasian		48 (77.5)	
Black and Brown-skinned		8 (13)	
Asian		6 (10)	
Education			
Elementary School		3 (5)	
Middle School		4 (6.5)	
High School		14 (23)	
Higher Education		41 (66)	
# Income Bracket			
1 to 2 × Minimum Wage		10 (16)	
2 to 5 × Minimum Wage		28 (45)	
6 to 9 × Minimum Wage		10 (16)	
10 + Minimum Wage		14 (23)	
Regular Physical Activity			
Yes		34 (55)	
No		28 (45)	
Anti-diabetic Medications		N	
Biguanides		20	
Sulfonylureas		20	
Dipeptidyl Peptidase 4 (DPP-4) Inhibitors		9	
Biguanides and Sulfonylureas in Association		15	
Other		35	
Cardiovascular Medications		N	
ACE Inhibitors and Calcium-channel Blockers		7	
Statins		18	
Beta-blockers		13	
ACE Inhibitors		9	
Diuretics		11	
Others		12	
	Average (SD)	Minimum	Maximum
Age (years)	69.8 (3.9)	65	79
Time Since Diagnosis (years)	15.1 (7.7)	2	38
Medicines (n)	3.6 (1.9)	1	8
Other Diseases (n)	1.4 (12)	0	5

Legend: SD—standard deviation; ACE—angiotensin-converting enzyme; n—number. # The minimum wage in Brazil corresponds to USD 282.40 per month, as of 14 October 2024.

**Table 2 ijerph-22-00184-t002:** Comparison of handgrip strength and isokinetic evaluation of knee extensors and flexors before and after 12 weeks of resistance training in older adults with T2DM, São Paulo, Brazil, 2021–2023, N = 62.

Test	Pre M (SD)	Post M (SD)	*p*-Value
Handgrip Test			
HGS (Dominant) (kg/f)	37.6 (8.0)	38.1 (7.3)	0.381
HGS (Non-dominant) (kg/f)	35.0 (8.1)	36.1 (7.4)	0.033 *
Isokinetic Evaluation			
Quadriceps (Dominant)			
PT/BW (%)	153.1 (32.2)	164.8 (32.0)	*p* < 0.001 *
TW(J)	515.0 (125.5)	534.2 (121.5)	0.003 *
Quadriceps (Non-dominant)			
PT/BW (%)	151.8 (31.0)	157.6 (37.0)	*p* < 0.001 *
TW(J)	509.8 (122.3)	538.0 (123.3)	0.007 *
Hamstrings (Dominant)			
PT/BW (%)	75.9 (19.9)	88.5 (20.4)	*p* < 0.001 *
TW(J)	285.6 (83.0)	316.4 (77.0)	*p* < 0.001 *
Hamstrings (Non-dominant)			
PT/BW (%)	71.2 (18.5)	82.9 (18.7)	*p* < 0.001 *
TW(J)	269.5 (78.5)	296.2 (75.6)	*p* < 0.001 *

Legend: HGS—handgrip strength; PT/BW—peak torque corrected for body weight; TW—total work, M—mean; SD—standard deviation. * *p* < 0.05.

**Table 3 ijerph-22-00184-t003:** Comparison of dynamic balance before and after 12 weeks of resistance training in older adults with T2DM, São Paulo, Brazil, 2021–2023, N = 62.

Test	Pre M (SD)	Post M (SD)	*p*-Value
Step/Quick Turn			
Turn Time (Non-dominant) (s)	2.4 (0.8)	2.1 (0.5)	0.111
Turn Time (Dominant) (s)	2.2 (0.8)	2.2 (0.6)	0.851
Turn Sway (Non-dominant) (°/s)	46.9 (13.2)	41.3 (14.8)	0.183
Turn Sway (Dominant) (°/s)	45.1 (14.8)	42.9 (14.7)	0.578
Step Up/Over			
Lift Up (Non-dominant) (%)	40.4 (17.0)	38.0 (10.1)	0.487
Lift Up (Dominant) (%)	34.9 (8.3)	38.9 (10.2)	0.051
Movement Time (Non-dominant) (s)	1.71 (0.3)	1.7 (0.3)	0.846
Movement Time (Dominant) (s)	1.78 (0.3)	1.71 (0.3)	0.558
Impact (Non-dominant) (%)	44.0 (12.6)	41.2 (12.4)	0.317
Impact (Dominant) (%)	43.6 (14.8)	39.6 (18.6)	0.339
Sit to Stand			
Weight Transfer (%)	0.52 (0.2)	0.50 (0.2)	0.802
COG Sway (º/s)	3.9 (1.2)	3.9 (1.5)	0.810

Legend: WT—weight transfer; COG—center of gravity.

## Data Availability

The authors may provide the data with request and co-participation in new analysis for scientific production.

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
