# Peer review of "Can 12-Week Resistance Training Improve Muscle Strength, Dynamic Balance and the Metabolic Profile in Older Adults with Type 2 Diabetes Mellitus?"

_ijerph, 2025, doi:10.3390/ijerph22020184_

Round 1
Reviewer 1 Report (Previous Reviewer 2)
Comments and Suggestions for Authors
I do not more have comment.
Author Response
We sincerely appreciate the thorough reviews provided, as they undoubtedly enhance the quality and rigor of our study. We have addressed the reviewers' comments directly within the PDF document, making it easier to verify the requested changes. Additionally, one of our co-authors, Dr. Catherine, a native English speaker, has meticulously revised the language to ensure clarity and precision.
Thank you for your attention and consideration.
Sincerely,
Angelica Castilho Alonso, Guilherme Carlos Brech and collaborators
Reviewer 2 Report (Previous Reviewer 1)
Comments and Suggestions for Authors
In this paper, older individuals with type II diabetes go through a 12 week resistance training regime to see if circulating measures of glucose utilization and biochemical markers of cardiovascular health are altered by the training regime. In the current study, training did result in increases in strength but not in measures of glucose, glucose utilization, or metabolic markers of health. In general, the paper has some interesting information but there are a number of edits that would make the manuscript easier to understand (please see the attached file) and more details need to be provided regarding the methods used to measure balance.
Some additional comments include:
1) The authors talk about how resistance exercise improves balance in the introduction but then they hypothesize that it has no effect. This does not make sense.
2) The authors acknowledge that there was no control group. Therefore, it is difficult to know what would have happened to people that did not train over a 12 week period. It is also possible that 12 weeks was not a long enough training period to increase strength but not improve measures of glucose utilization or biochemical measures of health.
3) Can authors please add information to the methods regarding; instructions to subjects on performing tests.
4) Did subjects exercise at a particular time during the day?

The English needs some editing. Please see comments in the attached file
Author Response
We sincerely appreciate the thorough reviews provided, as they undoubtedly enhance the quality and rigor of our study. We have addressed the reviewers' comments directly within the PDF document, making it easier to verify the requested changes. Additionally, one of our co-authors, Dr. Catherine, a native English speaker, has meticulously revised the language to ensure clarity and precision.
Thank you for your attention and consideration.
Sincerely,
Angelica Castilho Alonso, Guilherme Carlos Brech and collaborators

Reviewer 3 Report (New Reviewer)
Comments and Suggestions for Authors
Dear authors,
All my feedback is in the comment boxes within the attached manuscript PDF.
Overall the research study has been written up straightforwardly. As per the feedback provided, the introduction needs to elaborate further on previous similar research has been conducted and how your research is different.
With regard to the methodology, more information needs to be provided and I have questions regarding how and why only a limited range of upper-body muscular fitness (strength and power) was measured. I also fail to see the link between the intervention and handgrip strength being a measure of change over time.
For the results section, please check the level of significance for the results, For one or two of the results, I struggle to see the link.
The discussion and most importantly the limitations need to be attended to as per the comments in the PDF.
I look forward to reading the updated manuscript.
Kindest regards

Author Response
We sincerely appreciate the thorough reviews provided, as they undoubtedly enhance the quality and rigor of our study. We have addressed the reviewers' comments directly within the PDF document, making it easier to verify the requested changes. Additionally, one of our co-authors, Dr. Catherine, a native English speaker, has meticulously revised the language to ensure clarity and precision.
Thank you for your attention and consideration.
Sincerely,
Angelica Castilho Alonso, Guilherme Carlos Brech and collaborators

Round 2
Reviewer 3 Report (New Reviewer)
Comments and Suggestions for Authors
Dear authors,
The changes are most satisfactory. I do ask that the first sentence of the conclusion is changed, by removing the words "In conclusion" as they are unnecessary, especially when the heading is "Conclusion".
Kindest regards
Author Response
Thank you for your review.
we have removed the word in conclusion
best regards

This manuscript is a resubmission of an earlier submission. The following is a list of the peer review reports and author responses from that submission.
Round 1
Reviewer 1 Report
Comments and Suggestions for Authors
This paper looked at the ability of resistance training to reduce blood glucose and lipid measures and increase strength and muscle mass in men with controlled type II diabetes. This study was well designed, but the methods and results need some editing. Please see comments below.
Abstract
1) Line 16: “pre and post RT”
2) Line 15-17: the author should consider making the sentence that starts with “This is an experimental….” Into 2 sentences.
Introduction:
1) Page 2 line 67. The authors may want to add more detail to the sentence beginning “All these pathophysiological…”. For example consider, “Physiological changes associated with, or preceding the development of type II diabetes make individuals more susceptible to…..”
2) Page 2 last paragraph, can the authors provide a reference for the first sentence of that paragraph (The alarming prevalence…).
3) Page 2 sentence lines 79-80, what are non-consensual guidelines.
4) Page 2-3 last sentence, consider editing to say: Nonetheless, mechanisms underlying strength and balance gains during various types of RT are still to be elucidated; for example, the specificity, volume, and intensity of each exercise needs to be determined to achieve….”
5) Page 3 line 88. This should not be a new paragraph. This sentence can be part of the previous paragraph.
Methods
1) Subjects, line 114, do the authors mean “The primary outcome (or change with RT) was a change in Hb1Ac with an…” What about other outcomes such as increases in strength etc?
2) Page 4 line 134: Do the authors mean the average of 3 measurements/hand?
3) Page 4 lie 142, please describe how the maximum was determined for subjects.
4) Page 5 line 70, it is clear that blood was collected at baseline, but what other times was blood collected (i.e., what are post-interventional moments)?
5) Can the authors please provide a brief description of the OMNI scale? How did this compare to the weight they were asked to lift (e.g. were they lifting at 50 or 75% of maximal effort).
6) Statistics section please edit “The data were analyzed by comparing data from participants pre and post-intervention using a paired t- test or Wilcoxon test when the data were not normally distributed.
Results:
1) Page 6, Table 1. There are some things still written in Portuguese (?) for example for income bracket 1 a 2 should be 1 to 2. The authors should also consider providing some information or changing the income bracket so readers understand if participants had a lower or higher income.
2) In drugs used, outros needs to be changed to others.
3) Line 223, please edit to say “There were no significant differences in dynamic balance variables….”
4) Figures 1 and 2 are a bit confusing. The p value was mentioned in the analysis section, it does not need to be mentioned in every graphs (if the authors want the p value mentioned, it can be put in the figure caption).
a. In the first figure (HbA1c), please edit the y-axis to say % change from baseline.
Discussion
1) Page 10, what are the authors referring to in the first sentence (line 240) when they talk about progressive training.
2) The sentence beginning on line 241 with “Whereas….” is not a complete sentence.
3) Page 10, Line 234. The authors already defined T2DM, they do not need to do it again here.
4) A reduction of muscle mass of 100% would mean that a female has no muscle mass. This seems very unlikely.
5) Lines 247-248. Delete “Despite this” and start the sentence “The current study…” and add RT at the end of the sentence to clarify what was responsible for blocking muscle loss etc.
6) Please deleted the word “However” at the beginning of the sentence in line 251.
7) Are the authors still discussing grip strength in the paragraph that begins on page 11 (line 261)? If so, this can be added to the previous paragraph.
8) The paragraph beginning on line 273 can be combined with the previous paragraph. In addition, this paragraph talks about declines in balance in older adults. It’s difficult to say that the exercise prevented changes in balance. There wasn’t a no-exercise control. The study was conducted over a 12 week period, is there evidence that balance declines that quickly (over a 12 week period)?
9) Line 280, the authors talk about other factors that can contribute to changes in balance in people with type 2 diabetes. Did the authors ask if any subjects had peripheral neuropathy?
10) Line 293, please change evaluations to measures
Comments on the Quality of English Language
Please see above.
Author Response
Please find the attachment for detail.

Reviewer 2 Report
Comments and Suggestions for Authors
Thank you for submitting to IJERPH.
The authors' efforts in recruiting participants, intervention, data analysis, and writing the article are excellent. However, this study is not suitable for publication unless the analysis is re-done.
1. Many studies have already published similar results on resistance training. Therefore, the topic lacks freshness.
2. The analysis of this study is too monotonous. Furthermore, pre- and post-intervention comparison of a single group cannot be said to have discovered new scientific facts. It would be a more novel topic if the 62 participants were divided into two groups and the results were shown according to time and group. For example, a comparison of variables in the group with improved strength vs. the group with no improved strength could be a meaningful analysis.
The current design is not suitable for publication unless a new comparative analysis is conducted. Therefore, the interpretation of the results should be newly organized in the introduction, statistics, results writing, and discussion.
3. The completeness of the document is very low. There is 0.381 and .003*. There is no uniformity. The p in the table should be lowercase.
4. There is a lot of information that is not important in this study. Distinction between dominant and non-dominant, Covid and Vaccinated
Author Response
Please find the attachment for detail.

Reviewer 3 Report
Comments and Suggestions for Authors
André Luiz de Seixas Soares, Guilherme Carlos Brech, Adriana Machado-Lima, Joselma Rodrigues dos Santos, Julia Maria D’ Andréa Greve, Marcus Vinicius Grecco, Mara Afonso, Juliana Cristina Sousa , Ariana Tito Rodrigues, Matheus Henrique dos Santos Lino, Vanderlei Carneiro da Silva, Patricia Nemara Freitas de Souza Carneiro, Alexandre Lopes Evangelista, Catherine L. Davis, Angelica Castilho Alonso presented the work entitled "Can 12-week resistance training improve muscle strength, dy-2 namic balance and metabolic profile in older adults with type 2 3 diabetes mellitus? " Where authors analyzed the overall effect of a 12-week resistance training program on blood profiling and general physical parameters, resulting in enhanced functional responses without an improvement in the metabolic profile. While the reported results appear relevant, there are significant concerns that need to be addressed. Overall, the manuscript seems to be in its early stages, containing multiple self-notes and unfinished highlights, tables that lack detailed legends, and result section is incomplete. Additionally, the manuscript would benefit from scientific and grammatical improvements. Below are some examples of the major concerns with the current version of the manuscript:
Major Comments
Introduction
1) While the introduction provides valuable information and context, it lacks both a clear rationale and hypothesis. Specifically, the authors need to explain why they conducted this study.
Methods
2) Table 1 is unclear. It may be helpful to rearrange it by separating the "yes" and "no" variables into distinct columns. Information about individual participants might also need to be included as supplementary information.
3) While the exercises performed by patients are well detailed, there are no specifics on the clinical and or biochemical assays conducted (e.g., kit brand and CAS numbers, volumes, and sample storage specifications, if applicable).
4) Supplementary tables containing the metabolic test information for each patient need to be included.
5) Any information on the diet of patients can be of relevance due authors analyzed the lipid profiles
Results
6) The manuscript lacks a comprehensive results section beyond the presented graphs and brief descriptions. The results section needs more context to help the reader understand the figures.
Minor Comments
Discussion
7) The discussion might benefit from including some lines about how other types of exercise affect the profiles analyzed. For instance, does aerobic exercise affect the lipid profile in these patients? If so, why does resistance training not have the same effect?
Author Response
Please find the attachment for detail.
